# Comparison of Cell Fusions Induced by Influenza Virus and SARS-CoV-2

**DOI:** 10.3390/ijms23137365

**Published:** 2022-07-01

**Authors:** Chuyuan Zhang, Xinjie Meng, Hanjun Zhao

**Affiliations:** 1Department of Microbiology, School of Clinical Medicine, Li Ka Shing Faculty of Medicine, The University of Hong Kong, Hong Kong, China; zcymay@connect.hku.hk (C.Z.); xinjiem@hku.hk (X.M.); 2Centre for Virology, Vaccinology and Therapeutics, Hong Kong Science and Technology Park, Hong Kong, China; 3State Key Laboratory of Emerging Infectious Diseases, School of Clinical Medicine, Li Ka Shing Faculty of Medicine, The University of Hong Kong, Hong Kong, China

**Keywords:** cell fusions, fusion inhibitors, influenza virus, SARS-CoV-2

## Abstract

Virus–cell fusion is the key step for viral infection in host cells. Studies on virus binding and fusion with host cells are important for understanding the virus–host interaction and viral pathogenesis for the discovery of antiviral drugs. In this review, we focus on the virus–cell fusions induced by the two major pandemic viruses, including the influenza virus and SARS-CoV-2. We further compare the cell fusions induced by the influenza virus and SARS-CoV-2, especially the pH-dependent fusion of the influenza virus and the fusion of SARS-CoV-2 in the type-II transmembrane serine protease 2 negative (TMPRSS2^-^) cells with syncytia formation. Finally, we present the development of drugs used against SARA-CoV-2 and the influenza virus through the discovery of anti-fusion drugs and the prevention of pandemic respiratory viruses.

## 1. Introduction

Respiratory infectious diseases will be one of the major public health challenges in the future. During the past century, more than four influenza pandemics with millions of estimated deaths have emerged [1], and during the past twenty years, three coronavirus outbreaks, including SARS-CoV in 2003 [2], Middle East respiratory syndrome coronavirus (MERS-CoV) in 2012 [3] and SARS-CoV-2 in 2019, have emerged. Besides these outbreaks, the seasonal influenza viruses can cause more than 250,000 estimated global deaths every year [4]. In particular, SARS-CoV-2, which caused the COVID-19 pandemic, has continued for more than two years, leading to more than 0.5 billion infectious cases, with over 6 million deaths. The pandemic influenza virus, SARS-CoV-2 and seasonal influenza viruses have created severe social and economic burdens that have significantly affected our daily lives. The most important step for viral infection is virus–cell fusion, which allows viral genome release into the cytoplasm to start viral replication. The fusion process can occur either on the surface of the cell or within the endosome, as determined by viral factors of the fusion proteins and host factors including proteinases, receptors and attachment factors. Knowledge of the viral fusion process is important for preventing viral infectious diseases.

Currently, there are considered to be at least three classes of viral fusion proteins [5]. Class I fusion proteins (including influenza HA and the coronavirus spike protein) have a characteristic post-fusion conformation with a signature trimer of α-helical hairpins with a central coiled coil. Class II fusion proteins, such as the dengue E glycoprotein, have a structural signature of β-sheets forming an elongated ectodomain that refolds to produce a trimer of hairpins without the central coiled coil. Class III fusion proteins, such as the rabies virus G glycoprotein, combine the structural signatures of α-helical domains and β-sheet domains found in classes I and II proteins. The fusion-associated small transmembrane proteins, encoded by nonenveloped reoviruses, are regarded as the extra Class IV proteins, which are too small to form hairpins with a coiled-coil structure [6]. Here, we mainly review the cell fusions triggered by the influenza virus and SARS-CoV-2, the comparison between the cell fusions induced by the influenza virus and SARS-CoV-2, the potential drugs targeting viral fusion as the therapeutic treatment and the prevention of pandemic respiratory viruses.

## 2. Fusion Mediated by Influenza Virus

Influenza virus entry is mediated by the hemagglutinin (HA) protein, which is responsible for host–cell attachment and for membrane fusion between the viral envelope and the endosomal membranes. HA0 is a homologous trimer protein that requires a cleavage process during virus maturation in infected cells to split it into two subunits: HA1, which is important for binding to the cell receptors (sialic acids linked to galactose via an α-2,3 or an α-2,6 bond) [7], and HA2, which promotes membrane fusion [8]. Among the 18 subtypes of influenza HA, the highly pathogenic avian influenza A viruses of subtype H5 and H7 possess a multi-basic cleavage site that is cleaved by ubiquitously expressed proteases (including trypsin-like proteases, subtilisin-like endoprotease furin and serine proteases), allowing the systemic infection of these viruses in animals and humans [9]. In contrast, the HA0 of other influenza A and influenza B viruses (including A(H1N1)pdm09, H2N2 and others) possesses a monobasic cleavage site, which can be recognized only by a few trypsin-like proteases. These proteases are present in a restricted number of tissues, such as the respiratory tract or the intestinal tract, thus limiting the spread of viral infection [10].

Membrane fusion induced by the influenza virus begins with the binding of HA1 to the receptor sialic acids on the surface of the host cell. Binding is a necessary precursor step for bringing HA into the fusogenic conformation. Studies showed that blocking the viral binding with sialyllactose could destroy fusion at various pH values [11]. One hypothesis is that the energy pattern may change during the binding process, which can transform the fusogenic conformation state into a more stable energy state, in this way ensuring that fusion is triggered only if the virus infects the right target [12].

Following receptor binding and the uptake of the virus by the endocytosis, HA trimers are activated during the drop of pH in the maturing endosomes. Endosomal acidification triggers the separation of HA1 and the conformational change of HA2, forming a pre-hairpin structure and exposing the hydrophobic fusion peptide to the target membrane [11,13]. When the endosomal membrane approaches, the hydrophobic fusion peptide is inserted into the interior of the membrane and is anchored there through strong hydrophobic interactions [14]. After this insertion step, several HAs form a cluster to overcome the strong hydrophobic force between the viral membrane and the endosomal membrane. It is generally believed that the number of HA trimers in this cluster should be greater than one, which is confirmed by cryo-electron microscopy images [13,15].

Next, the proteins within the fusion unit undergo a further conformational change, bending at a hinge point to bring the two opposing membranes together and dehydrating the space between them. The two outermost leaflets of the membrane merge to form a “stalk”, where hemifusion takes place, the lipids mixing from the outer layer [16].

Over time, either the stalk or a nearby segment of the membrane ruptures, creating a fusion pore through which the viral genome can escape into the cytoplasm of the host cells [13]. It has also been suggested that, under some conditions, influenza virus HA initiates fusion by puncturing one of the contact membranes, forming a leaky “rupture-insert” structure that promotes hemifusion [17]. The mechanism for the expanding of fusion pores requires further exploration.

## 3. SARS-CoV-2-Mediated Fusion

### 3.1. Two Fusion Pathways of SARS-CoV-2

SARS-CoV-2 can enter cells through two distinct pathways: the plasma membrane fusion pathway mediated by TMPRSS2 and the cathepsin-mediated viral membrane, and the endosomal membrane fusion pathway [18]. The monomeric spike (S) protein of SARS-CoV-2 is the viral transmembrane protein that can directly mediate SARS-CoV-2 entry into host cells and belongs to the Class I fusion proteins exemplified by the influenza HA protein. Similar to other Class I fusion proteins, the S protein is synthesized as a precursor that is subsequently cleaved and then initially folded into the metastable trimeric form and undergoes major irreversible conformational changes after a certain triggering to induce fusion reaction.

Unlike SARS-CoV, the multiple basic amino acids located at the S1–S2 junction of the S protein of SAR-CoV-2 are easily cleaved into S1 and S2 subunits by furin during the maturation of the virus in an infected cell. Therefore, the S protein on the mature virion consists of two non-covalently linked subunits: the S1 subunit binding to the angiotensin-converting enzyme 2 (ACE2), and the S2 subunit anchoring to the membrane. After binding to ACE2, the conformational change of the S1 subunit exposes the S2′ cleavage site. During the direct plasma membrane entry, the S2 protein is processed by TMPRSS2 to trigger the fusion. If the expression of TMPRSS2 is insufficient and the spike–ACE2 complex cannot encounter the protease, the virus can enter cells through clathrin-mediated endocytosis. In the endocytic pathway, endo-lysosome proteases, such as cathepsin L, which require an acidic environment for their activity, perform the S2′ cleavage to trigger the fusion [19].

### 3.2. Fusion Triggered by SARS-CoV-2

In these two entry pathways, the cleavage of the S2′ site can expose the fusion peptide and initiate a cascade of refolding events in S2, including the dissociation of S1 from S2, which can propel the fusion protein forward into the target membrane. Finally, the conformational changes to the post-fusion structure of S2 bring the two membranes together, facilitating fusion–pore formation, which enables the membrane fusion between the virus and the host cells [20].

SARS-CoV-2-mediated cell fusion events and their consequences, including syncytia formation [21] and cell–cell transmission [22], have been widely reported. Studies on the S protein mainly use the former phenomenon as the indicator of cell–cell fusion, while the latter is not reliant only on the plasma membrane fusion [23]. The interaction of the S protein with the ACE2 receptor on the cell surface with the expression of TMPRSS2 can trigger the plasma membrane fusion. After entry, the viral genome is released into the cytoplasm for replication. After RNA transcription, the S protein is processed by furin and other proteases during transportation from the endoplasmic reticulum (ER) to Golgi compartments through coat protein I (COPI) and COPII vesicles. The sequestered S protein is packaged into virions, and the virions exit the cell via deacidified lysosome-dependent exocytosis, while the leaked S protein can appear on the cell surface and induce the fusion [21].

In the case of SARS-CoV, the interaction of membrane proteins helps to retain the S protein on intercellular membranes. The COPI binding motifs at the c-terminus of S protein promote the interaction. [24]. Similarly, the lack of the C-terminal moiety on the S protein or the lack of the co-expression envelope (E) and membrane (M) proteins of SARS-CoV-2 could induce the accumulation of the S protein on the cell surface and aggravate syncytia formation in a cell culture model [25]. Another study suggested that the COPI binding site of the SARS-CoV-2 S protein was sub-optimal and could enhance the ability to form syncytia [26].

Studies have highlighted the significance of the S1/S2 furin cleavage site in SARS-CoV-2 for inducing syncytia formation [27]. The optimization of the S1/S2 cleavage site could facilitate cell fusion, and other studies have suggested that the removal of the furin cleavage site substantially reduced, but did not block, infectivity and cell fusion [28,29]. Natural mutations in these amino acid sites can affect syncytia formation, such as D614G or P681R [30,31,32].

## 4. The Comparison of Cell Fusions Mediated by the Influenza Virus and SARS-CoV-2 

### 4.1. The pH-Dependent Fusion of Influenza Virus

Although both the influenza virus and the coronavirus enter host cells through membrane fusion mediated by Class I fusion proteins, the triggering events of the fusion proteins are different. The relatively simple trigger of influenza virus HA is the proton binding when the endosomal pH is lowering, but cell fusion triggered by the SARS-CoV-2 spike does not rely directly on the low pH in endosomes (Figure 1). The low pH-dependent HA-mediated fusion can take place mainly in the endosomes. No syncytia formation takes place when no artificial challenge is introduced by treating the cells with the low pH [33]. However, SARS-CoV-2 spike-ACE2 binding can induce the syncytia in the normal condition of the cell culture without additional artificial treatment, such as the low pH challenge. This difference may be an important factor in the capacity of the SARS-CoV-2 infection to cause severe syncytia in the lungs of patents, but influenza virus infection has not been reported as a cause of obvious syncytia in patients. The fusion activity of SARS-CoV-2 in cells and the syncytia formation are more likely correlated with the severe clinical outcomes in COVD-19 patients, such as the inflammatory responses in patients with severe COVID-19 [34,35].

### 4.2. Protease-Dependent Fusion of SARS-CoV-2

After the cleavage of the HA1–HA2 junction and the binding to the host receptors, HA2 of the influenza virus is activated mainly by the low pH in the endosomes to trigger the conformation change and the fusion process. However, after the cleavage of S1–S2 junction, S2 conformation changes of the SARS-CoV-2 spike are not sensitive to pH. When undergoing the first step of the conformation change induced by receptor binding, the SARS-CoV-2 spike exposes the S2′ site to the next step of activation by the host proteases, including the cell surface serine proteases (including TMPRSS2, TMPRSS13 and others) and the endosomal proteases (cathepsin B/L), which require low pH in the endosomes for activation [36]. The cleavage of the S2 site by these host enzymes could expose the fusion peptide of the spike to trigger the fusion of the virus membrane with the plasma membrane or the endosomal membrane. Studies showed that using different protease inhibitors targeting TMPRSS2 or cathepsin L could effectively inhibit SARS-CoV-2 entry or spike-mediated fusion, which demonstrates that the spike-mediated fusion depends on the S2 activation by these enzymes.

### 4.3. Syncytia Formation Induced by SARS-CoV-2 Infection in TMPRSS2^-^ Cells

The syncytia formation mediated by the spike in TMPRSS2 positive (TMPRSS2^+^) cells is well understood, and is mainly due to the direct plasma membrane fusion between cells expressing ACE2 and cells expressing the spike activated by TMPRSS2 (Figure 1). However, in TMPRSS2^-^ cells (293T or VeroE6), the spike of SARS-CoV-2 could efficiently trigger the cell fusion and form syncytia [18,30]. Multiple syncytia were found in severe pneumonia cases and might be related to pathogenesis and severe inflammation responses [35,37,38]. The cell fusion in TMPRSS2^-^ cells should be less likely to take place on the plasma membrane because there is no TMPRSS2 to activate S2 to expose the fusion peptide to trigger the plasma membrane fusion. How does the fusion mediated by spike-ACE2 occur in TMPRSS2^-^ 293T cells or VeroE6 cells? Currently, there is no clear evidence to support the direct cell–cell fusion on the surface of the plasma membrane when there is no TMPRSS2 in 293T or VeroE6 cells. Therefore, it is supposed that the fusion is induced by the endocytic pathway, which allows the activation of S2 with the endosomal proteases (Figure 1). This possibility has been supported by using endosomal acidification inhibitors, including chloroquine and bafilomycin A1, to inhibit the spike-ACE2-mediated cell fusion in TMPRSS2^-^ 293T cells [18]. Moreover, interferon induced transmembrane protein 2 (IFITM2) and IFITM3 located in the endo-lysosomal compartments could effectively restrict the cell fusion mediated by the spike of SARS-CoV-2 variants in TMPRSS2^-^ VeroE6 cells, but not in TMPRSS2^+^ cells [30]. These studies suggested that the syncytia formation of the spike-mediated fusion in TMPRSS2^-^ cells most likely needs to go through the endocytic pathway, but does not take place mainly on the cell surface (Figure 1). More studies are warranted to further investigate the syncytia formation mediated by the spike of SARS-CoV-2 in TMPRSS2^-^ cells.

### 4.4. Ca^2+^-Dependent Fusion of the Influenza Virus and SARS-CoV-2

Virus-mediated cell fusion can be modulated by other host factors. Fusions induced by influenza and SARS-CoV-2 are Ca^2+^ dependent. Influenza A virus infection induces oscillations in the cytosolic Ca^2+^ concentration of host cells, the prevention of which markedly attenuates virus internalization and infection, mainly by modulating endocytosis [39]. The expression of the spike protein can affect the intracellular Ca^2+^ fluctuations and cause the increased expression of the TMEM16F ion channel and scramblase, which can translocate phosphatidylserine from the cytofacial leaflet of the plasma membrane to the exofacial leaflet, promoting the fusogenic properties of the membrane spike protein. The exposure of phosphatidylserine is associated with a variety of virus–cell and cell–cell fusion events [35]. Moreover, the extent of the membrane binding of the SARS-CoV-2 fusion protein is significantly reduced in the absence of Ca^2+^ ions, suggesting that its fusion is Ca^2+^-dependent [40].

## 5. Drug Development by Inhibiting the Viral Fusion

### 5.1. Targeting the Endocytic Pathway against the Influenza Virus and SARS-CoV-2

Both the influenza virus and the coronavirus (CoV) can enter the host cells through receptor-mediated endocytosis, and the virus–cell fusions take place in the late endosomes. Thus, endosomal trafficking or acidification inhibitors, which are antivirals with broad-spectrum antiviral activities, might be useful in fighting against pandemics, especially for influenza viruses and SARS-CoV-2 with high mutation rates (Table 1). For example, the disruption of the AP2-associated protein kinase 1 (AAK1), which regulates the process of endocytosis, can block the viral entry and fusion [41]. Baricitinib, a Janus kinase (JAK) inhibitor inhibiting AAK1, was suggested for trial on patients with SARS-CoV-2 infection [42] and has shown clinical benefits in COVID-19 patients [43,44]. The endosomal acidification inhibitors are known to suppress pH-dependent viral replication. Bafilomycin A1, the inhibitor of vacuolar-type H+-ATPase (V-ATPase) to block pH decrease in the endosomes, can exhibit broad-spectrum antiviral properties in vitro and in vivo [45]. Also, inhibitors targeting the endosomal proteases (E-64d or MDL28170, cysteine protease inhibitors) could effectively inhibit the influenza virus and the SARS-CoV-2 infection [46,47]. However, the off-target side effects are problems for clinical use. Another well-known antiviral candidate, chloroquine, is the endosomal acidification inhibitor that could inhibit pH-dependent viruses like the influenza virus and coronavirus in some in vitro studies, but its effect in vivo needs to be investigated further [48,49].

The off-target side effects of these host-targeting inhibitors preventing host endosomal acidification might be the main barrier for their use as therapeutic drugs. The antivirals inhibiting virus- and host-endosomal acidification can improve the antiviral efficacy by specifically targeting the virus, which could show better antiviral efficacy against the influenza virus and SARS-CoV-2 in vivo when compared with chloroquine [18,50,51].

**Table 1 ijms-23-07365-t001:** Viral fusion inhibitors with broad-spectrum activities against viruses in vivo.

Antivirals	Broadly Targeting	Mechanism of Blocking Viral Fusion	Status (In Vitro, In Vivo or In Trial)
Arbidol	CoV and influenza virus	Block conformation changes and endosomal components to affect viral fusion	In clinical trial [52,53]
Bafilomycin A1	CoV and influenza virus	V-ATPase inhibitor preventing endosomal acidification to block viral fusion	In vivo [45]
Camostat	CoV and influenza virus	Target TMPRSS2 to interfere with viral entry and fusion	In clinical trial [54]
Chloroquine	CoV and influenza virus	Increase endosomal pH to block viral fusion	In clinical trial [55]
E-64d	CoV and influenza virus	Cathepsin B/L inhibitor blocking viral entry and fusion through the endocytic pathway	In vivo [56]
LJ001	CoV and influenza virus	Block fusion by affecting membrane flexibility	In vivo [57]
Niclosamide	Adv, CoV, influenza virus, rhinovirus and RSV	Multifunctional drug blocks fusion and endosomal acidification	In clinical trial [58]
Nitazoxanide	CoV, influenza virus and parainfluenza virus	Protein disulfide isomerase inhibitor affects viral fusion	In clinical trial [59]
Nafamostat mesylate	CoV and influenza virus	Serine protease inhibitor blocking viral fusion	In clinical trial [60]
P9, P9R, 8P9R	CoV, influenza virus and rhinovirus	Multifunctional peptides block fusion by clustering viral particles and inhibiting endosomal acidification	In vivo [18,50,51]
Tyrphostin A9	Influenza virus, Sendai virus and murine CoV	Tyrosine kinase inhibitor blocks clathrin-mediated viral entry and fusion	In vivo [61]

### 5.2. Targeting Cleavage Sites

The inhibition of influenza A virus through the targeting of the HA0 (HA precursor) cleavage could be achieved by using inhibitors on furin, such as phenylacetyl-Arg-Val-Arg-4-amidinobenzylamide [62,63]. However, for SARS-CoV-2, the furin cleavage of S1/S2 promotes but is not essential for infection and cell–cell fusion, suggesting that furin inhibitors may reduce but do not abolish viral spread [28,29]. Since TMPRSS2 is involved in many viral infections, such as coronavirus and influenza, it would be an attractive target for use against a wide spectrum of respiratory viruses (Table 1). For example, camostat mesylate (a serine protease inhibitor) could block TMPRSS2 binding to the spike protein in vitro [47], but the in vivo antiviral activity against the influenza virus and SARS-CoV-2 has not been well approved. Nafamostat mesylate, another serine protease inhibitor used for treating disseminated intravascular coagulation (DIC), was found to possess an even higher inhibitory capacity for the entry of the SARS-CoV-2 virus into host cells and showed clinical benefits in COVID-19 patients [64]. In addition, tyrosine kinase inhibitor (tyrphostin A9) showed broad-spectrum antiviral activities in vitro and in vivo against the influenza virus, the coronavirus and other viruses, probably by interfering with the clathrin-mediated viral entry and fusion [61]. However, these host-targeting antivirals for inhibiting the influenza virus and SARS-CoV-2 will need to be studied further to investigate the antiviral efficacy and the side effects in vivo.

### 5.3. Preventing Conformation Changes

After binding to host receptors, the influenza HA or SARS-CoV-2 spike proteins require conformational changes to trigger the membrane fusion. Due to their similar structure, the conformational changes of fusion proteins have some commonalities. Class I fusion proteins can form a coiled-coil six-helix bundle (6HB) structure in the post-fusion state, which is considered as a conserved antiviral target. Using this strategy, studies are able to design relatively broad-spectrum antiviral lipopeptides against both the influenza virus and the coronavirus [65], specifically for the influenza virus by targeting the HA stalk region [33], and specifically for the coronavirus including SARS-CoV-2 by targeting the heptad repeat 1 (HR1) region of the spike protein [66].

Small molecule compounds binding to viral fusion proteins can also affect the viral protein conformation changes (Table 1). Arbidol can inhibit the influenza virus by targeting HA to block the conformation change [67]. Recent studies have indicated that it blocks the entry and intracellular trafficking of SARS-CoV-2, which is likely due to the prevention of membrane fusion [18,68]. However, its clinical effect seems to be unstable in different trials, perhaps because of the rapid metabolism and the low serum concentration [69]. In addition, the conformational changes of the viral proteins might require the rearrangement of intramolecular disulfide bonds, catalyzed by protein disulfide isomerase (PDI) [70]. Nitazoxanide, as a potential PDI inhibitor, has been shown to broadly inhibit the influenza virus, parainfluenza virus, coronavirus and other viruses with an unclear mechanism [71].

### 5.4. Targeting Host Membrane Components

Lipids represent the structural foundations of cellular and viral membranes, which play an important role in viral infection. Lipid-lowering drugs such as fibrates, targeting fatty acid synthesis and increasing lipoprotein lipase activity, could increase the survival of mice infected with the influenza virus [72]. Lipid-lowering drugs such as statins and PCSK9 inhibitors were also suggested to be beneficial in COVID-19 treatment due to their significant effect on the prevention of cardiovascular disease and their anti-inflammatory effect [73]. Bavituximab (PGN401) represents one of the clinically promising broad-spectrum anti-viral paradigms. It targets the anionic phospholipid phosphatidylserine, which can be exposed on the cell surface during apoptotic events that can be triggered by viral infection [70]. In addition, some broad-spectrum antivirals targeting lipid oxidation were considered as hopeful candidates for fighting against new pandemics, such as Rhodanine derivative LJ001, which was found to inhibit coronavirus replication after viral entry in vitro and in vivo [74,75] (Table 1). However, their anti-viral mechanism can be complex, with the exception of the anti-fusion effect. More evidence is needed for the development of these anti-lipid drugs as broad-spectrum therapeutic antivirals.

Besides directly targeting lipids, a broad-spectrum approach against enveloped viruses may be made possible by targeting the hydrophobic surfaces on the membranes, such as peptides that interact with the hydrophobic membrane–protein interfaces [76,77]. Many antiviral peptides (AVPs) whose therapeutic utilities have been validated have been collected in the AVPdb database, including ones used against the influenza virus and the coronavirus [78].

In addition, calcium ion regulation is also important in membrane fusion. Relative drugs, including niclosamide and nitazoxanide, which can block Ca^2+^ release, could inhibit cell fusion triggered by the spike-ACE2 binding [79]. These drugs showed broad antiviral activities against the influenza virus, coronavirus, adenovirus, respiratory syncytial virus (RSV) and rhinovirus [80,81]. More studies are needed to investigate the underlying mechanism of inhibition and the potential side effects in vivo.

## 6. Prevention for Pandemic Respiratory Viruses

SARS-CoV-2 and the pandemic influenza virus can spread around the world in a very short time. So far, there has been no effective method for predicting and preventing the emergence and transmission of new viruses to humans. High mutation rates and gene recombination provide opportunities for viruses to acquire important site mutations, or antigenic drift or shift, thereby increasing viral fitness in humans and possibly also increasing the likelihood of the emergence of new viruses and resistance to drugs or antibodies induced by vaccination. Thus, broad-spectrum antivirals can play important roles in preventing novel emerging viruses.

For the early-stage prevention of new emerging virus outbreaks, drug screening of approved antivirals with broad antiviral activity, or new combinations of current drugs, may be good options for the rapid development of antivirals to fight against epidemics and to reduce morbidity and mortality. Moreover, many biological targets in the process of membrane fusion have not been fully studied and utilized. As biopharmaceuticals mature, as exemplified by the development of AVPs and compound libraries, targeting viral fusion with broad-spectrum antiviral activity is a promising innovation direction for the prevention of new emerging viruses. For the late-stage prevention of new emerging virus outbreaks, we know from the mortality and severe cases during the SARS-CoV-2 pandemic that high vaccination rates are important in preventing pandemic respiratory viruses.

Aside from the discovery of new broad-spectrum antivirals and widespread vaccination, wearing surgical masks, keeping hands clean and less contact with wild animals are also suggested to be important factors in reducing the transmission and emergence of new viral pathogens in humans.

## Figures and Tables

**Figure 1 ijms-23-07365-f001:**
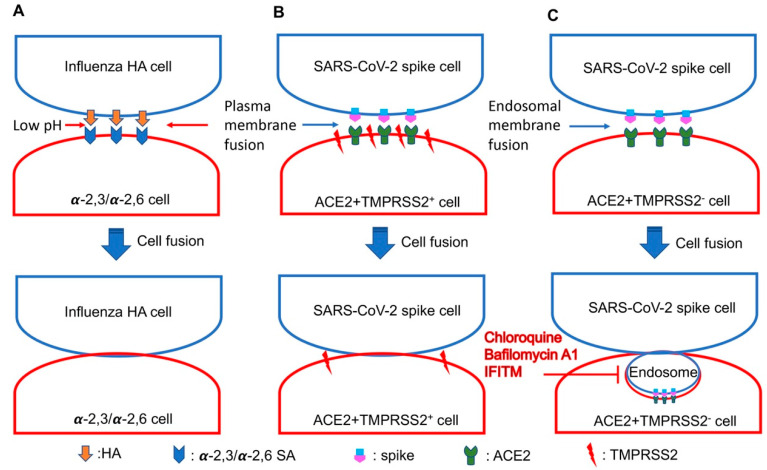
The schematic model of HA- and spike-mediated cell fusions. (**A**) Influenza HA binds to sialic acids linked to galactose via an α-2,3 bond or α-2,6 bond (α-2,3/α-2,6 SA). Influenza virus HA can trigger cell–cell membrane fusion after the low pH challenge. (**B**) SARS-CoV-2 spike binds to ACE2 to trigger cell–cell plasma membrane fusion when there is TMPRSS2 on the cell membrane. (**C**) SARS-CoV-2 spike binds to ACE2 to trigger cell–cell membrane fusion through the endocytic pathway when there is no TMPRSS2 on the cell membrane. The two adjacent cell membranes with spike-ACE2 binding can bend to undergo endocytosis, which can provide the low pH in the endosomes to allow cathepsin B/L to cleave the S2 site to trigger the fusion. This kind of fusion can be blocked by endosomal inhibitors (chloroquine, bafilomycin A1 or IFITM).

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
