# Peer review of "Comparison of Cell Fusions Induced by Influenza Virus and SARS-CoV-2"

_ijms, 2022, doi:10.3390/ijms23137365_

Round 1

Reviewer 1 Report

This review is well prepared and brings comprehensive data that can be useful for the development of a new strategy of anti-covid and anti-influenza therapy. Unfortunately, the chapter about the therapeutic manipulation of cell membrane fusion is a little confusing for the reader. I strongly suggest summarising data in a table where should be noted the name of the substance, the mechanism of action, and the system in which it was tested (in vitro, type of cell, in vivo, clinical study....).  

Author Response

Review 1:

This review is well prepared and brings comprehensive data that can be useful for the development of a new strategy of anti-covid and anti-influenza therapy. Unfortunately, the chapter about the therapeutic manipulation of cell membrane fusion is a little confusing for the reader. I strongly suggest summarising data in a table where should be noted the name of the substance, the mechanism of action, and the system in which it was tested (in vitro, type of cell, in vivo, clinical study....).  

Response:

Thank you for your suggestion. We now summarized the fusion inhibitors against influenza virus and SARS-CoV-2 in the new table 1.

Reviewer 2 Report

The review is interesting and well-written. The Authors have scientific experiences on CoVs and in this manuscript they focused on the virus-cell fusions induced by the two major pandemic viruses, influenza virus and SARS-CoV-2.

Few minor suggestions are listed in order to improve the work. References section should be improved and the Authors must better clarify the schematic model of spike-mediated cell fusions of SARS-CoV-2 spike that binds to ACE2 to trigger cell-cell plasma membrane fusion when there is noTMPRSS2 on cell membrane. Moreover, the suggestions for preventing pandemic respiratory viruses must be better described and discussed. Paragraph 6 looks like a further comment on the drug development against SARS-CoV-2 and influenza virus by discovering anti-fusion drugs.

Check for spelling and punctuation errors.

Author Response

Review 2:

The review is interesting and well-written. The Authors have scientific experiences on CoVs and in this manuscript they focused on the virus-cell fusions induced by the two major pandemic viruses, influenza virus and SARS-CoV-2.

Few minor suggestions are listed in order to improve the work. References section should be improved and the Authors must better clarify the schematic model of spike-mediated cell fusions of SARS-CoV-2 spike that binds to ACE2 to trigger cell-cell plasma membrane fusion when there is noTMPRSS2 on cell membrane. Moreover, the suggestions for preventing pandemic respiratory viruses must be better described and discussed. Paragraph 6 looks like a further comment on the drug development against SARS-CoV-2 and influenza virus by discovering anti-fusion drugs.

Check for spelling and punctuation errors.

Response:

Thank you for the comments. We cited some new references and revised the reference section.

We described the cell-cell fusion in cells without TMPRSS2 in the figure legend with more detailed information (from line 204 to 206).

We revised the paragraph of prevention for pandemic respiratory viruses and checked the spelling and language carefully.